# Relationship between Reticulorumen Parameters Measured in Real Time and Methane Emission and Heat Stress Risk in Dairy Cows

**DOI:** 10.3390/ani12233257

**Published:** 2022-11-23

**Authors:** Ramūnas Antanaitis, Lina Anskienė, Eglė Rapaliutė, Ronaldas Bilskis, Karina Džermeikaitė, Dovilė Bačėninaitė, Violeta Juškienė, Remigijus Juška, Edita Meškinytė

**Affiliations:** 1Large Animal Clinic, Veterinary Academy, Lithuanian University of Health Sciences, Tilžės St. 18, LT-47181 Kaunas, Lithuania; 2Department of Animal Breeding, Veterinary Academy, Lithuanian University of Health Sciences, Tilžės Str. 18, LT-47181 Kaunas, Lithuania; 3AUGA Group, AB, Konstitucijos pr. 21C, LT-08130 Vilnius, Lithuania; 4Department of Ecology, Animal Science Institute, Lithuanian University of Health Sciences, R. Zebenkos 12, 82317 Baisogala, Lithuania; 5Animal Husbandry Selections, Breeding Values and Dissemination Center, Agriculture Academy, Vytautas Magnus University, Universiteto St. 10A, Akademija, LT-53361 Kaunas, Lithuania

**Keywords:** precision farming, greenhouse gas emissions, global warming

## Abstract

**Simple Summary:**

We have found only a few publications in the literature on the link between rumen pH and CH_4_. Based on this, we hypothesized that reticulorumen pH and temperature, where the latter is registered on an online system, affect greenhouse gas CH_4_ emissions. As a result, to the best of our knowledge, this is the first study that has assessed the relationship of CH_4_ emissions with reticulorumen pH and temperature. According to the aim of this study, we found that cows with a higher pH (6.22–6.42) produce 46.18% more methane emissions than cows with a lower pH. Moreover, cows with a higher risk of heat stress had a higher risk of subclinical acidosis. The novel aspect of the study is that, by using real-time reticulorumen pH, a temperature-registration system, and a laser methane detector, we could establish a relationship between reticulorumen parameters measured in real time and methane emissions and heat-stress risk in dairy cows. For this reason, more studies should be conducted to evaluate this process.

**Abstract:**

The objective of this study was to investigate a connection between CH_4_ emissions and reticulorumen pH and temperature. During the experiment, we registered the following parameters: reticulorumen pH (pH), reticulorumen temperature (RR temp.), reticulorumen temperature without drinking cycles, ambient temperature, ambient relative humidity, cow activity, heat index, temperature–humidity index (THI), and methane emissions (CH_4_). The experimental animals were divided into two groups based on the reticulorumen pH: 1. pH < 6.22 and 2. pH 6.22–6.42. We found that cows assigned to the second pH class had higher (46.18%) average values for methane emissions (*p* < 0.01). For the other indicators, higher average values were detected in cows of the first pH class, RR temperature (2.80%), relative humidity (20.96%), temperature–humidity index (2.47%) (*p* < 0.01), and temperature (3.93%) (*p* < 0.05), which were higher compared to cows of the second pH class. Reticulorumen pH was highly negatively correlated with THI and temperature (r = −0.667 to 0.717, *p* < 0.001) and somewhat negatively with heat index, relative humidity, and RR temperature (r = −0.536, *p* < 0.001; r = −0.471 to 0.456, *p* < 0.01). Cows with a higher risk of heat stress had a higher risk of lower reticulorumen pH.

## 1. Introduction

Globally, emissions from farmed ruminants account for roughly 20 percent of agricultural emissions [1]. Agriculture is a significant source of GHG emissions which are released into the atmosphere, accounting for roughly 30% of total anthropogenic emissions, including indirect emissions through land-cover change. The three main greenhouse gases released by animal production are carbon dioxide (CO_2_), CH_4_, and nitrous oxide (N_2_O) [2,3]. Animal husbandry is a significant source of greenhouse gas emissions, accounting for 14.5 percent of global emissions, approximately the same as the transportation sector [4,5]. Globally, ruminant animals are anticipated to release between 80 and 95 million tons of CH_4_ each year [3]. The enteric fermentation process is responsible for more than 90 percent of animal CH_4_ emissions and 40 percent of agricultural greenhouse gas emissions [4,6].

Methane is produced in the rumen as a result of microbial fermentation. Reduced CH_4_ emissions will halt climate change and lower greenhouse gas levels [7]. Increasing milk yield lowers climate consequences at the animal and farm levels [8]. The new findings could stimulate more research into the effects of methanogenesis suppression on rumen fermentation and post-absorptive metabolism, which could increase animal productivity and efficiency [7].

According to Cantor [9], reticulorumen temperature is an effective predictor of aspects of cattle well-being, such as daily herd water intake or inflammation [9]. Alzahal et al. [8] investigated the association between ruminal temperature and pH, as well as their potential to predict nutritional and health status in dairy cows. Cooper-Prado et al. [9] found that the ruminal temperature reduces one day before parturition. A ruminal pH of 6.0 to 6.4 promotes optimal food fermentation and fiber absorption. Cellulolytic bacteria digest fiber effectively at this pH level, which is inhibited at pH values lower than 6.0 [10]. As a result, a decrease in ruminal pH increases acidity, which increases abomasum temperature [11]. The reticuloruminal temperature monitoring clearly reflects the cows’ feed- and water-consumption habits [12].

Ruminal pH regulates rumen physiology and fermentation in a variety of ways, including methanogenesis [13]. Ruminal pH is regulated by interactions between organic acid production from microbial feed fermentation, bicarbonate input into the rumen via saliva, secretion via the ruminal epithelium, SCFA absorption and transit, and, perhaps, ammonia absorption [14]. The optimal ruminal pH range for methanogen growth is pH 6.0 to 7.5, with the quickest growth rate occurring at near-neutral pH, while a drop in ruminal pH leads to a slower rate of methanogen growth and less activity [13]. Van Kessel and Russell [13] discovered that when the pH of rumen fluid from forage-fed cows was decreased to 6.0, in vitro CH_4_ generation stopped. Diets heavy in soluble carbohydrates or starch can cause the ruminal pH to remain at 5.5–6.0 for long periods of time. Despite the fact that pH 5.5–6.0 has been reported to be adequate to restrict CH_4_ generation in vitro, Hünerberg et al. [15] demonstrated that even pH levels indicative of SARA (5.5) and ARA (5.2) had little effect on CH_4_ generation in vivo. As a result, lowering ruminal pH is not a viable CH_4_-mitigation method, and is not the primary determinant of the reduction in CH_4_ (grams per kilogram DMI) associated with high-grain diets. Modifications in methanotrophic community structure toward more pH-tolerant strains, or sequestration of methanogens within microenvironments with higher pH levels than rumen fluid (e.g., increased ecto- and endosymbiosis with protozoa), could be mechanisms that allow methanogens to adapt to low ruminal pH conditions [15].

Continuous rumen pH and temperature monitoring could be useful for assessing the impact of water temperature on ruminal parameters in cattle [16,17]. At the moment, a bolus inserted into the rumen can measure a cow’s temperature in real time (reticulorumen). The boluses can measure both temperature and pH. Wireless boluses can transmit data every ten minutes. The data can be saved in the cloud or on a PC. Measurements can be taken for up to a year, depending on the battery life of the various bolus versions [18]. The sensor is positioned in the reticulorumen, where it is influenced by fermentation heat, which is 0.5 degrees Celsius greater than body temperature and the transient cooling impact of the cow’s drinking water [18]. Recent advancements in automated animal monitoring technology have demonstrated potential for monitoring heat stress in cattle, while taking individual behavioral and activity profiles into account [19]. Future platforms for autonomous monitoring and mitigation of heat stress in cattle are likely to be based on minimally invasive smart technologies, either individually or as part of an integrated system, allowing for real-time solutions to animal responses in a variety of production systems and environmental conditions [19].

According to our previous research, the interline recorded pH of the cow reticulum can be utilized to predict the animal’s health and reproductive status [20]. High-yielding Holstein cows generated the most milk, resulting in high enteric methane emissions during the early lactation stage, but when milk production was taken into consideration, late-lactating cows were the greatest contributors to methane emissions (CH_4_ intensity expressed per kg of energy-corrected milk yield) [21].

In the literature, we found only a few studies on the link between rumen pH and CH_4_. Based on this, we hypothesized that reticulorumen pH and temperature, where the latter is registered on an online system, affects GHG emissions. According to this hypothesis, the aim of this study was to establish a relationship between GHG emissions and reticulorumen pH and temperature.

## 2. Materials and Methods

*Farm and Animals.* This study was carried out in accordance with the provisions of the Lithuanian Law on Animal Welfare and Protection. The study’s approval number is PK016965. The experiment was carried out at the Lithuanian University of Health Sciences using 650 milking Holstein cows (55.792368° N, 24.017499° E) in one of Lithuania’s dairy farms, from April to August 2022. The study was conducted on clinically healthy cows in their second lactation, with an average daily milk yield of 32.19 ± 1.05 kg per cow, average feed intake of 18 kg DM/day, milk fat of 4.25 (±0.25), milk protein of 3.45 (±0.15), milk somatic cell count of 180,000/mL (±0.55), and milk urea nitrogen of 25% (±5). Cows were kept in a free stall barn and milked using a DeLaval milking parlor.

Data on individual attributes were gathered from the farm’s computer system and recorded on a spreadsheet (lactation number, breed, latest calving date, and milk yield) (Delpro DeLaval Inc, Tumba, Sweden). The number of days in milk (DIM) for each cow was determined for each data collection period by calculating the number of days between the last calving date and the first day of the data collection period. The cows were allowed unrestricted access to the feeding table. All cows were fed a total mixture ration (TMR), with maize and alfalfa silage as the principal forages. TMR contained a mixture of grass silage (38%), corn silage (38%), and flaked grain concentrate with mineral mixture (24%). The ration was designed to satisfy or surpass the needs of a 550 kg Holstein cow that produces 35 kg of milk each day, depending on the situation (Table 1). Cows were fed every day at 08:00 a.m. and 04:00 p.m.

### 2.1. Experimental Design

#### 2.1.1. Measurements

During the experiment, we registered the following parameters: reticulorumen pH (pH), reticulorumen temperature (RR temp.), reticulorumen temperature without drinking cycles, ambient temperature, ambient relative humidity, cow activity, heat index, temperature–humidity index (THI), and methane emissions (CH_4_) (Table 2).

#### 2.1.2. Classification of Animals

The experimental animals were separated into two groups based on the reticulorumen pH assay: 1. pH < 6.22 (*n* = 25, 69.0% of cows) and 2. pH 6.22–6.42 (*n* = 11, 31.0% of cows). Classes were assigned according to our previous publication [22]. Moretti et al. defined six THI classes for analysis: safe (68), moderate pain (68 THI 72), discomfort (72 THI 75), alert (75 THI 79), danger (79 THI 84), and emergency (84). No pain or emergency THI levels were observed [23].

The two groups of animals were separated on the basis of registered parameters from April to August 2022. During this period, conditions changed only for the following parameters: heat index, relative humidity, temperature, and THI.

### 2.2. Measurement of Reticulorumen pH, Temperature, and Walking Activity

It was possible to monitor real-time parameters such as pH, temperature of reticulorumen content (TRR), and cow activity by smaXtec boluses (smaXtec animal care technology^®^) [24,25]. Antennas (smaXtec animal care technology^®^) were used to collect the data. The pH, TRR, and activity of the animals were monitored using an indwelling and wireless data-transmitting system (smaXtec animal care GmbH, Graz, Austria). A microprocessor was in charge of controlling the system. Data on pH and TRR were collected using an analog-to-digital converter (A/D converter) and stored in an external memory chip for later analysis and interpretation. To begin the experiment, the pH probes were calibrated with buffer solutions of pH 4 and pH 7 in order to ensure that they worked properly. All of the information was gathered through the use of the smaXtec messenger^®^ computer software.

### 2.3. Measurement of CH_4_

The laser methane detector was designed to detect gas leaks from a safe distance in gas transmission networks, landfills, and other sites where there is a possibility of CH_4_ leakage [26]. The LMD has various advantages, including its flexibility, portability, and ease of use. It also does not require an external power supply. As a result, it is reasonably inexpensive to employ in a wide range of experimental and commercial situations [27]. Researchers have further developed and evaluated the measurement, refined the analysis of data obtained with the LMD, and applied the LMD in studies on genetic analyses [28], [29], nutrition and feed efficiency [30,31], and the physiological status of animals [29], as well as to characterize different husbandry systems [32]. Chagunda et al. [26] found sensitivity and specificity of 95.4% and 96.5%, respectively, for cows, and sensitivity and specificity of 93.8% and 78.7%, respectively, for sheep.

The measurement with the handheld LMD HESAI HS4000 (Hesai, Building L2-B, Hongqiao World Centre, Shanghai) is based on infrared absorption spectroscopy. By detecting a fraction of the diffusely reflected laser beam, the integrated CH_4_ concentration between the LMD and the target is determined [33]. The measured value is represented as CH_4_ column density (ppm), which is the sum of the CH_4_ concentrations along the laser route, or the average CH_4_ concentration (ppm) multiplied by the path length (m). The LMD measures CH_4_ from 0.5 to 50,000 ppm m (up to 5 vol%), and can be utilized at a distance of 0.5 to 30 m. The data are displayed in real time on the LMD’s display, and an audio-visual warning is issued if a particular threshold is surpassed. Following the physiology of the cow, the gas excreted directly from the rumen (eructation) is first inhaled into the lungs and then exhaled again with each respiratory cycle. LMD is aimed at the area around the animal’s nostrils, which is the main point source of emitted CH_4_. The measurements are performed by the same operator during the study period. An operator holds the LMD by hand and follows the animal’s head movements. Chagunda et al. [26] were the first to use the LMD to measure the concentration of CH_4_ in the breath of dairy cows. They used the LMD at a distance of 3 m from the cow, and took recordings at the nostrils for 15–25 s at a time.

Measurement intervals were 0.5–1 s (i.e., one or two CH_4_ values per second) [34]. Measurements were performed in an experiment at the same time of day, that is, 2 h after feeding. During the study period, all of the measurements were recorded at the same time of day, approaching cows in a similar way, at the same distance, and with the same LMD angle [27].

### 2.4. Data Analysis and Statistics

For the statistical analysis, we used version 25.0 of IBM SPSS 25.0 Statistics for Windows (IBM Corp., Armonk, NY, USA). Using descriptive statistics, normal distributions of variables were assessed using the Kolmogorov–Smirnov test. The results were produced as the mean and standard error (M ± SE). Mean differences between groups were analyzed using the Student’s *t*-test. The Pearson correlation coefficient was calculated in order to define the linear relationship between the investigated variables. A linear regression equation was calculated to determine the statistical relationship between methane CH_4_ (dependent variable) and date (independent variable) during each week (05/13, 06/08, 06/15, 06/22). A probability below 0.05 was considered to be reliable (*p* < 0.05).

## 3. Results

Data analysis of our research revealed that cows assigned to the second pH class (pH 6.22–6.42) had higher (46.18%) average values of methane emissions (Table 3), *p* < 0.01, while for the other indicators, higher average values were detected in cows of the first pH class (pH < 6.22). RR temperature (2.80%), relative humidity (20.96%), temperature–humidity index (2.47%), (*p* < 0.01) and temperature (3.93%) (*p* < 0.05) were higher compared to cows of the second pH class (pH 6.22–6.42).

Reticulorumen pH was highly negatively corelated with THI and temperature (r = −0.667 to 0.717, *p* < 0.001) and somewhat negatively with heat index, relative humidity, and RR temperature (r = −0.536, *p* < 0.001; r = −0.471 to 0.456, *p* < 0.01) (Figure 1).

Methane was weakly negatively corelated with heat index (r = −0.341, *p* < 0.05), and weakly positively with RR pH (r = 0.370, *p* < 0.05) (Figure 2).

For the temperature–humidity index, we found a highly positive statistically significant relationship with temperature (r = 0.995, *p* < 0.001), a somewhat positive relationship with relative humidity (r = 0.598, *p* < 0.001) and a highly negative relationship with RR pH (r = −0.717, *p* < 0.001) (Figure 3).

Statistically significant mean differences in methane emissions between hours were detected only on 13 May, when emissions were 61.40% higher (*p* < 0.01) at 10:00 a.m. and 44.98% higher (*p* < 0.05) at 09:00 a.m., compared to methane emissions at 11:00 a.m. Analysis of our data revealed that methane emissions increased during each week of the study (09:00 a.m. y = 36.868x + 113.81, R^2^ = 0.8171; 11:00 a.m. y = 39.565x + 88.795, R^2^ = 0.633), except for during the last week of the experiment; at 10:00 a.m. on 22 June, methane emissions decreased (y = −2.999x + 237.11 R^2^ = 0.025). The analysis revealed statistically significant mean differences at 09:00 a.m. between 13 May–22 June, being 41.15% higher on 22 June (*p* < 0.01). The lowest methane emissions were detected at 11:00 a.m. on 13 June, when they were 58.79% lower compared to 15 June (*p* < 0.01), 58.67% lower compared to 22 June (*p* < 0.05), and 57.64% lower compared to 8 June (*p* < 0.001) (Table 4).

## 4. Discussion

We aimed to determine the relationship between GHG emissions and reticulorumen pH and temperature. Wireless telemetry technologies have been used to develop boluses for the purpose of monitoring the pH of the rumen, and are now frequently used to control/measure physiological parameters that can indicate livestock diseases, including the detection of subacute ruminal acidosis and other variations in the ruminal environment. As a result, this technique increases timely diagnosis of illness and improves the protection of animal health and productivity [16,17]. Continuous administration of reticulorumen boluses, as in the current investigation, allows for prompt responsiveness to changes in the animal’s condition. Therefore, the novel aspect of this study is the application of reticulorumen boluses for real-time monitoring and lowering heat stress, and this can potentially be applied to dairy cattle [18].

By using real-time reticulorumen pH, a temperature registration system, and a laser methane detector, we found that cows with a reticulorumen pH of 6.22–6.42 had 46.18% higher average methane emissions. Lana et al. [35] found that ruminal pH impacted ruminal methane emissions. Ruminal pH regulates rumen physiology and fermentation in a variety of ways, including methanogenesis [36]. It is regulated by interactions between organic acid generation from microbial fermentation of feed, bicarbonate flow into the rumen through saliva, secretion via the ruminal epithelium, and possibly ammonia absorption [37]. The optimal ruminal pH range for methanogen growth is pH 6.0 to 7.5, with the highest growth rate of this microbe occurring at a pH around neutral, while a dip in ruminal pH leads to a slower rate of methanogen growth and less activity [36].

Although pH 6 has been reported to be sufficient to suppress CH_4_ generation in vitro, some investigation shows that even pH levels indicative of subclinical acidosis (5.5) and clinical acidosis (5.2) do not decrease CH_4_ production in vivo. As a result, lowering ruminal pH is not a viable CH_4_-mitigation method, and is not the primary determination factor in the reduction in CH_4_ (grams per kilogram DMI) associated with high-grain diets. Changes in the methanogen community structure toward more pH-tolerant strains, or sequestration of methanogens within microenvironments with higher pH than rumen fluid (e.g., greater ecto- and endosymbiosis with protozoa), could allow methanogens to adapt to low ruminal pH settings [15]. Changes in community structure toward more pH-tolerant strains, as well as sequestration within microenvironments in biofilms or protozoa where methanogens are protected from low pH, could allow methanogens to survive bouts of low ruminal pH [15]. When fed concentrates, the rumen has a low pH and an elevated propionate proportion, both of which have been linked to reduced CH_4_ production [35]. In our research, using real-time measured reticulorumen parameters, we found that cows with a reticulorumen pH of 6.22–6.42 had 46.18% higher average methane emissions.

We found that methane emissions increased for each hour in each week of the study (09:00 a.m. y = 36.868x + 113.81, R^2^ = 0.8171; 11:00 a.m. y = 39.565x + 88.795, R^2^ = 0.633), except in the last week of the experiment at 10:00 a.m., when methane emissions decreased (y = −2.999x + 237.11 R^2^ = 0.025).

It is intriguing that a decrease in ruminal pH did not correspond with a decrease in CH_4_ emissions, given that pH 6 completely reduced CH_4_ production in vitro [13,15].

With the exception of a slight drop 2.5 h later, methane levels in heifers fed with growth diets increased immediately after feeding and peaked 8 h later [15]. In the growing and finishing portion of the investigation, the rate of CH_4_ synthesis steadily decreased for dairy cows fed once per day with ad libitum intake, a result that likely represents the drop in fermentable organic matter in the rumen during the day [38]. The rate of CH_4_ emissions from dairy cows fed twice or four times per day fell more rapidly with each feeding [38]. Thus, in the growth assays, the increase in the CH_4_ emission rate immediately after feeding (5.94 g/h before feeding to 9.41 g/h 1 h after feeding) might be attributed to a rapid increase in dissolved H2 and its use as a substrate for methanogenesis [15]. During the transition to a diet containing 65 percent grain and 35 percent grass hay, the organization of the methanogen community in the rumen of nonlactating dairy cows fed with grass hay changed [39]. Although it is unclear whether the changes in methanogen consortia composition in response to high-grain diets are primarily due to a drop in rumen pH, the findings of Hook et al. [39] indicate that some rumen methanogen species are better adapted to low pH than others.

According to the results of this study, reticulorumen pH was highly negatively correlated with THI and temperature (r = −0.667 to 0.717, *p* < 0.001). Temperature and relative humidity are two environmental parameters that might influence animal feed consumption [40]. Long-term exposure of lactating dairy cows to high ambient temperature and relative humidity reduces their ability to disperse heat generated by both metabolic processes and heat acquired from the environment, thus making them vulnerable to heat stress [41]. Heat stress was observed to be associated with changes in ruminal bacterial composition and metabolites, with more lactate-producing species and less acetate-producing species in the population, potentially affecting milk production [42]. An increase in temperature and RH reduces animal dry matter intake and rumination due to an increase in the amount of buffering agents entering the rumen, which could be related to decreased chewing activity [43]. Zhao et al. [42] found that rumen pH and acetate concentrations were significantly lower in heat-stressed cows. Cows undergoing heat stress had greater ruminal temperature, and experienced a 5% drop in ruminal pH [44]. Ruminal changes may be linked to cow performance during heat stress [45]. Furthermore, blood flow is shifted from the gastrointestinal tract to the periphery for heat dissipation, which reduces digestibility even further [40]. Furthermore, increased respiration during the summer season raises expired CO_2_ output, resulting in lower blood and rumen pH, as well as acidosis [40]. Castro-Costa et al. [46] found a decrease in ruminal pH in heat-exposed ruminants, which they attributed to decreased rumen fermentation during heat stress.

## 5. Conclusions

Based on this study’s objective to discover a link between GHG emissions and reticulorumen pH and temperature, we can conclude that cows with a higher pH (6.22–6.42) produce 46.18% more methane emissions than cows with a lower pH do. Furthermore, cows with a higher risk of heat stress had a higher risk of subclinical acidosis. The novel aspect of this study is that, by using real-time reticulorumen pH, a temperature registration system, and a laser methane detector, we could establish a relationship between the reticulorumen parameters measured in real time and the methane emissions and heat stress risk in dairy cows. More studies should be conducted in order to evaluate this process.

## Figures and Tables

**Figure 1 animals-12-03257-f001:**
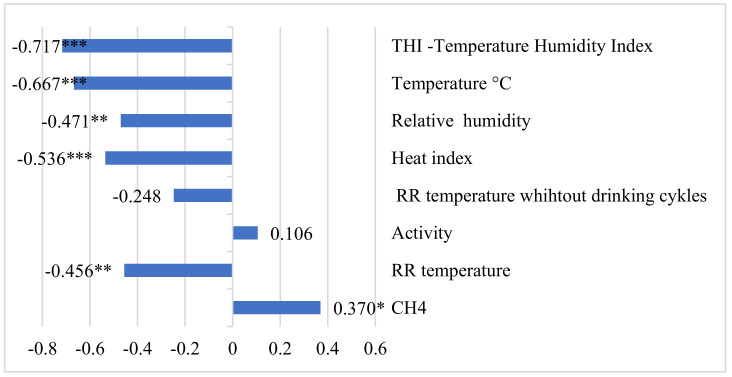
Correlation coefficients between reticulorumen pH, methane and reticulorumen indicators. * *p* < *0.05*, ** *p* < 0.01, *** *p* < 0.001. CH_4_—methane; RR—reticulorumen.

**Figure 2 animals-12-03257-f002:**
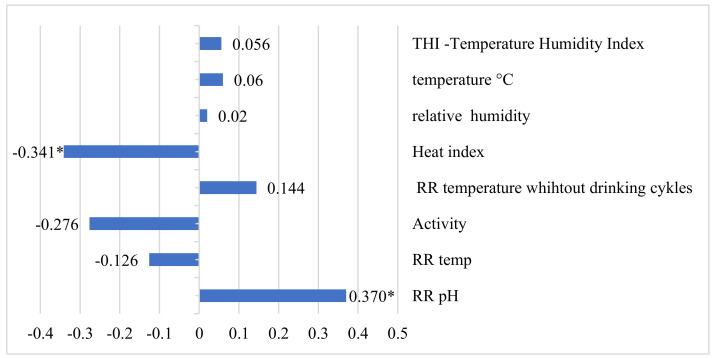
Correlation coefficients between methane, temperature humidity index, and reticulorumen indicators. * *p* < 0.05. CH_4_—methane; RR—reticulorumen.

**Figure 3 animals-12-03257-f003:**
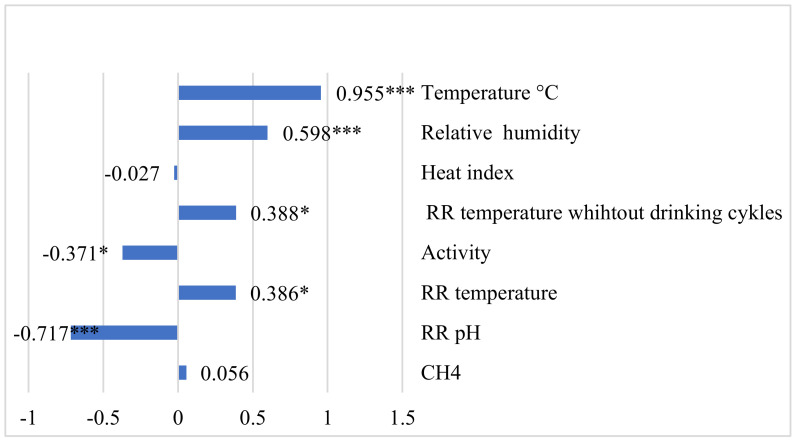
Correlation coefficients between THI, methane, and reticulorumen indicators. * *p* < 0.05, *** *p* < 0.001. CH_4_—methane; RR—reticulorumen.

**Table 1 animals-12-03257-t001:** Chemical composition of feed ration.

Nutrients	Composition
Dry matter (DM)	49%
Neutral detergent fiber	28 (% of DM)
Acid detergent fiber	20 (% of DM)
Crude protein	16 (% of DM)
Non-fiber carbohydrates Net energy for lactation	39 (% of DM) 1.80 Mcal/kg DM

DM—dry matter; % of DM—percent of dry matter; Mcal/kg—megacalories per kilogram.

**Table 2 animals-12-03257-t002:** Parameters and intervals of their measurements.

Parameter	Units	Interval of Measurements
Reticulorumen pH	Value	April–August 2022
Reticulorumen temperature	°C	April–August 2022
Reticulorumen temperature without drinking cycles	°C	April–August 2022
Ambient temperature	°C	April–August 2022
Ambient relative humidity	%	April–August 2022
Cow activity	Steps/hour	April–August 2022
Heat index	-	April–August 2022
Temperature–humidity index	-	April–August 2022
Methane emission	ppm	Estimated from multiple breath measurements with laser methane detector (LMD) HESAI HS4000 from April to August 2022 (at the same time: 09:00, 10:00, 11:00 a.m.). LMD recordings were 3–5 min

**Table 3 animals-12-03257-t003:** Means and standard errors of the investigated indicators based on the reticulorumen pH assay: 1. pH< 6.22 (*n* = 25, 69.0% of cows), 2. pH 6.22–6.42 (*n* = 11, 31.0% of cows). The letters ^a,b,^ represent statistically significant differences between classes. ** p < 0.05*, ** *p* < 0.01.

Ph Class	CH_4_ (ppm)	RR Temperature (°C)	Cow Activity (Steps/Hour)	RR Temperature without Drinking Cycles (°C)	Heat Index	Relative Humidity %	Ambient Temperature (°C)	THI
1. pH < 6.22	187.64 ± 20.33 ** ^b^	38.59 ± 0.16 ** ^b^	6.52 ± 0.83	39.46 ± 0.03	2.08 ± 0.60	70.95 ± 2.81 ** ^b^	17.82 ± 0.17 * ^b^	63.16 ± 0.32 ** ^b^
2. pH 6.22–6.42	348.64 ± 56.50 ** ^a^	37.51 ± 0.36 ** ^a^	6.30 ± 0.54	39.39 ± 0.02	1.60 ± 0.21	56.08 ± 4.91 ** ^a^	17.12 ± 0.11 * ^a^	61.60 ±0.03 ** ^a^

CH_4_—methane; RR—reticulorumen; THI—temperature–humidity index.

**Table 4 animals-12-03257-t004:** The analysis of methane emissions by hour and date from April to August 2022. The letters ^a, b,^ and ^c^ indicate statistically significant differences between hours; ^A, B, C,^ and ^D^ indicate statistically significant differences between dates. ** p <* 0.05, *** p* < 0.01, **** p* < 0.001.

	Date			
Hours	13/05 (A)	08/06 (B)	15/06 (C)	22/06 (D)
1. 9 (a)	166.28 ± 3.00 ^*c;^ ^**D^	177.62 ± 9.51	197.46 ± 24.48	282.56 ± 6.86 ^**A^
2. 10 (b)	237.05 ± 2.10 ^**c^	210.24 ± 4.38	261.01 ± 9.47	210.13 ± 4.08
3. 11 (c)	91.49 ± 6.34 ^**b; *a***B, **C, *D^	215.98 ± 2.29 ^***A^	221.99 ± 7.17 ^**A^	221.37 ± 5.03 ^*A^

## Data Availability

The data presented in this study are available within the article.

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
