# Peer review of "Relationship between Reticulorumen Parameters Measured in Real Time and Methane Emission and Heat Stress Risk in Dairy Cows"

_animals, 2022, doi:10.3390/ani12233257_

Round 1
Reviewer 1 Report
The manuscript named “Relation Of Reticulorumen Parameters And Temperature-Humidity index With Methane Emission In Dairy Cows” showed the application of reticulorumen boluses as useful tool for a real-time controlling methane emission and risk of heat stress. There is still some parts need to be improved before it published.
1. In the Introduction, please add more information or studies about reticulorumen pH and temperature on performance of cows, and give us the reason why you studied the rumen pH and CH4 connection, rather than just saying “In literature we found only a few publications about rumen pH and CH4 connection.”
2. In materials and methods, please give us more studies or information about the accuracy of this method to measure CH4.
3. In Results part, please replace Figure 4 with Table to make the results more clear for readers to review.
4. In Discussion part, please be concise and add information about the reason why you reach these results.
5. Use more recent reference.
Author Response
Dear Reviewer,
Authors are very thankful with the comments, which help us to improve the manuscript. All changes proposed have been included in the manuscript and highlighted in yellow and track changes.
Best Regards,
Prof. Ramunas Antanaitis
|
Question |
Answers |
|||||||||||||||||||||||||
|
In the Introduction, please add more information or studies about reticulorumen pH and temperature on performance of cows, and give us the reason why you studied the rumen pH and CH4 connection, rather than just saying “In literature we found only a few publications about rumen pH and CH4 connection.” |
In introduction section we added – “Methane is produced in the rumen as a result of manure degradation and microbial fermentation. Reduced CH4 emissions will halt climate change and lower greenhouse gas levels [7]. Increasing milk yield lowers climate consequences at the animal and farm levels [8]. The new findings could lead the way for more research into the effects of methanogenesis suppression on rumen fermentation and post-absorptive metabolism, which could increase animal productivity and efficiency [7]”
“Van Kessel and Russell [13] discovered that when the pH of rumen fluid from forage-fed cows was decreased to 6.0, in vitro CH4 generation stopped. Diets heavy in soluble carbohydrates or starch can cause the ruminal pH to stay at 5,5- 6.0 for long periods of time. Despite the fact that pH 5.5- 6.0 has been reported to be adequate to restrict CH4 generation in vitro..”
“Indeed, wireless telemetry technologies have been used to develop boluses for monitoring the pH of the rumen, and are now frequently used to control/measure physiological parameters that can indicate livestock diseases, such as the detection of subacute ruminal acidosis and other variations in the ruminal environment. As a result, this technique increases illness diagnosis timeliness and adds to the protection of animal health and productivity. Continuous rumen pH and temperature monitoring could be useful for assessing the impact of water temperature on ruminal parameters in cattle [16], [17]. At the moment, a bolus put into the rumen can measure a cow's temperature in real time (reticulorumen). The boluses can measure both temperature and pH. Wireless boluses can transmit data every ten minutes. The data can be saved in the cloud or on a PC. Measurements can be taken for up to a year depending on the battery life of the various bolus versions [18]. The sensor is positioned in the reticulorumen, where it is influenced by fermentation heat, which is 0.5 degrees Celsius greater than body temperature, and the transient cooling impact of the cow's drinking water [18]. Recent advancements in automated animal monitoring technology have showed promise in monitoring heat stress in cattle while taking individual behavioral and activity profiles into account [19]. Future platforms for autonomous monitoring and mitigation of heat stress in cattle are likely to be based on minimally invasive smart technologies, either individually or as part of an integrated system, allowing for real-time solutions to animal responses in a variety of production systems and environmental conditions [19]” “Based on this we hypothesized that reticulorumen pH and temperature, registered with on line system has impact on GHG emission. According to this hypothesis the aim of this study was to find a relation between GHG emission and reticulorumen pH and temperature”
|
|||||||||||||||||||||||||
|
2. In materials and methods, please give us more studies or information about the accuracy of this method to measure CH4. |
We added information and references – “The laser methane detector was designed to detect gas leaks from a safe distance in gas transmission networks, landfills, and other sites where there is a possibility of CH4 leakage [26]. Chagunda et al. [26] were the first to use the LMD to measure the concentration of CH4 in dairy cow breath. The LMD has various advantages, including its flexibility, portability, and ease of use. It also does not require an external power supply. As a result, it is reasonably inexpensive to employ in a wide range of experimental and commercial situations [27]. Researchers have further developed and evaluated the measurement, refined the analysis of data obtained with the LMD, and applied the LMD in studies on genetic analyses [28], [29], nutrition and feed efficiency [30], [31], the physiological status of animals [29], and to characterize different husbandry systems [32]. Chagunda et al.[26] found sensitivity and specificity of 95.4% and 96.5% for cows, respectively, and sensitivity and specificity of 93.8% and 78.7% for sheep. The measurement with the handheld LMD HESAI HS4000 (Hesai, Building L2-B, Hongqiao World Centre, Shanghai) is based on infrared absorption spectroscopy. By detecting a fraction of the diffusely reflected laser beam, the integrated CH4 concentration between the LMD and the target is determined [33]. The measured value is represented as CH4 column density (ppm), which is the sum of the CH4 concentrations along the laser route or the average CH4 concentration (ppm) multiplied by the path length (m). The LMD measures CH4 from 0.5 to 50,000 ppm m (up to 5 vol%) and can be utilized from a distance of 0.5 to 30 m. The data is displayed in real time on the LMD's display, and an audio and visual warning is issued if a particular threshold is surpassed. Following the physiology of the cow, the gas excreted directly from the rumen (eructation) is first inhaled into the lungs and then exhaled again with each respiratory cycle. LMD is aimed at the area around the animal’s nostrils, which is the main point source of emitted CH4. The measurements are performed by the same operator during the study period. An operator holds the LMD by hand and follows the animal’s head movements. Chagunda et al. [26] were the first to use the LMD to measure the concentration of CH4 in dairy cow breath. They used the LMD at a distance of 3 m from the cow and recorded for 15-25 seconds at a time at the nostrils. Distance of 3 m from the cow and taking recordings at the nostrils for 15–25 s at a time. Measurement intervals 0,5-1 s (i.e., one or two CH4 values per second) [34]. Measurements performed in an experiment at the same time of day – 2 hours after feeding. During the study period all the measurements performed same time of the day, approaching cows in the most similar way, same distance and keeping same LMD angle [27]”
|
|||||||||||||||||||||||||
|
3. In Results part, please replace Figure 4 with Table to make the results more clear for readers to review. |
We changed from figure 4 to table 4.
Table 4. Distribution of methane emission by hour and date. Different letters a, b, c indicate statistically significant differences between hours; A, B, C, D indicate statistically significant differences between date. * P < 0.05, ** P < 0.05, ** P < 0.01, *** P < 0.001).
|
|||||||||||||||||||||||||
|
4. In Discussion part, please be concise and add information about the reason why you reach these results. |
We corrected and added new section “discussion” |
|||||||||||||||||||||||||
|
5. Use more recent reference. |
Corrected |
Reviewer 2 Report
The topic of this communication is interesting. Also the potential of the boluses as a management tool is evident. However, I had a hard time to read the paper and interpretate the data presented.
Regarding the experimental design I get the impression the 2 groups of animals were kept in different locations and/or that the measuements were not performed at the same time. At least the ambient temp of the groups was significant different. This makes the experimental design weak. Furthermore I miss in the observations the feed intake and milk production/composition of the groups.
In the experimental design 2 groups are mentioned but in the figures presented it seems that it is analysed as one group. Did the correlations differ for both groups.
Feed intake is correlated with ambient temp, production level and rumen pH. Therfore it also affects the methane production.
The increase in methane production in the first period of the experiment was mentioned. How about the rest of the experimental period. ?
Minor remarks:
line 41 : This is not supported by the data. Only the correlation is reported. Not the change for SARA
line 78-79: at pH 6.0 should be a range fe 5,5- 6.0 . It would be surprising if it would be constant.
line 101: registered with on linesystem could be left out.
line 120: What is the % grass silage 40 %?.
Line 125: I do not understand what 39 % NE laction means
line 126 : Feed=fed
line 170-172: pH group 2. Although reference is given a short description of how the classes were determined would improve the readability.
line 179-180: What do you mean. It offers the possibility to take action
line 238: estimated ?
line 266-268: were are the diurnal data?
Author Response
Dear Reviewer,
Authors are very thankful with the comments, which help us to improve the manuscript. All changes proposed have been included in the manuscript and highlighted in yellow and track changes.
Best Regards,
Prof. Ramunas Antanaitis
|
Question |
Answers |
||||||||||||||||||||||||||||||||||||||||||||||||||||||||||||||||||||||||||||||||
|
Regarding the experimental design I get the impression the 2 groups of animals were kept in different locations and/or that the measuements were not performed at the same time. At least the ambient temp of the groups was significant different. This makes the experimental design weak. Furthermore I miss in the observations the feed intake and milk production/composition of the groups. |
We added information - The experiment was carried out at the Lithuanian University of Health Sciences, with 650 milking Holstein cows (55.792368°N, 24.017499° E) in one of Lithuania's dairy farms from April to August 2022. The study was conducted on clinically healthy cows in their second lactation (that had an average daily milk yield of 32.19 ± 1.05 kg per cow), average of feed intake - 18 kg DM/day, milk fat – 4.25 (±0.25) milk protein – 3.45 (±0.15), milk somatic cell count -180000/ml (±0.55), milk urea nitrogen – 25% (±5). Cows were kept in a free stall barn and milked using a Delaval milking parlour. We corrected table 2 – Table 2. Parameters and intervals of their measurements.
Also, we added – “The two groups of animals were created on the basis of registered parameters from April to August 2022. During this period, conditions changed only keeping condition (heat index, relative humidity, temperature, and THI)”
|
||||||||||||||||||||||||||||||||||||||||||||||||||||||||||||||||||||||||||||||||
|
In the experimental design 2 groups are mentioned but in the figures presented it seems that it is analysed as one group. Did the correlations differ for both groups. |
Yes, the experimental animals were divided into two groups based on the reticulorumen pH: 1. pH 6.22 (69.0% of cows), 2. (31.0% of cows). Two groups were investigated only for comparision of differences in groups of reticulorumen pH. Due to low amount of cows in a second group of reticulorumen pH ( n=11) we didn‘t made any calculations of correlations according to groups of reticulorumen pH. Correlation coefficients between investigated indicators according to 1 st group of RR Ph (n=25)
Correlation coefficients between investigated indicators according to 2 nd group of RR pH (n=11)
* P < 0.05, ** P < 0.01, *** P < 0.001. Should we use these tables with correlation coefficients between investigated indicators according to groups of RR Ph in a manuscript?
|
||||||||||||||||||||||||||||||||||||||||||||||||||||||||||||||||||||||||||||||||
|
Feed intake is correlated with ambient temp, production level and rumen pH. Therfore it also affects the methane production. |
We made calculations with feed intake. Correlation coefficients between feed intake and investigated indicators. * P < 0.05.
Feed intake was weakly positive related with milk yield (r=0.329, P < 0.05) and with ambient temperature and THI (from r=0.375 to r=0.353 , P < 0.05).
|
||||||||||||||||||||||||||||||||||||||||||||||||||||||||||||||||||||||||||||||||
|
The increase in methane production in the first period of the experiment was mentioned. How about the rest of the experimental period. ?
|
We found that methane emission increased each week of the study (9 a. m. y = 36,868x + 113,81, R² = 0,8171; 11 a.m. y = 39,565x + 88,795, R² = 0,633), except of 10 a. m., were methane emission decreased at last week of experiment (y = -2,999x + 237,11 R² = 0,025).
|
||||||||||||||||||||||||||||||||||||||||||||||||||||||||||||||||||||||||||||||||
|
Minor remarks: |
|
||||||||||||||||||||||||||||||||||||||||||||||||||||||||||||||||||||||||||||||||
|
line 41 : This is not supported by the data. Only the correlation is reported. Not the change for SARA |
We corrected to – “Cows with higher risk of heat stress had higher risk of lower reticulorumen pH” |
||||||||||||||||||||||||||||||||||||||||||||||||||||||||||||||||||||||||||||||||
|
line 78-79: at pH 6.0 should be a range fe 5,5- 6.0 . It would be surprising if it would be constant. |
We corrected to – “Diets heavy in soluble carbohydrates or starch can cause the ruminal pH to stay at 5,5- 6.0 for long periods of time. Despite the fact that pH 5.5- 6.0 has been reported to be adequate to restrict CH4 generation in vitro…” |
||||||||||||||||||||||||||||||||||||||||||||||||||||||||||||||||||||||||||||||||
|
line 101: registered with on linesystem could be left out.
|
We corrected to – “registered with on line system has impact on GHG emission. According to this hypothesis the aim of this study was to find a relation between GHG emission and reticulorumen pH and temperature” |
||||||||||||||||||||||||||||||||||||||||||||||||||||||||||||||||||||||||||||||||
|
line 120: What is the % grass silage 40 %?. |
We corrected to – “TMR contained a mixture of grass silage (38%), corn silage (38%), flaked grain concentrate with mineral mixture (24 %)” |
||||||||||||||||||||||||||||||||||||||||||||||||||||||||||||||||||||||||||||||||
|
Line 125: I do not understand what 39 % NE laction means |
|
||||||||||||||||||||||||||||||||||||||||||||||||||||||||||||||||||||||||||||||||
|
line 126 : Feed=fed |
We corrected to – “Cows were fed every day at 08:00 am and 04:00 pm.” |
||||||||||||||||||||||||||||||||||||||||||||||||||||||||||||||||||||||||||||||||
|
line 170-172: pH group 2. Although reference is given a short description of how the classes were determined would improve the readability. |
We have added the amound of cows in each group.
The experimental animals were separated into two groups based on the reticulorumen pH assay: 1. pH< 6.22(n=25, 69.0% of cows), 2. pH 6.22–6.42 (n=11, 31.0% of cows). |
||||||||||||||||||||||||||||||||||||||||||||||||||||||||||||||||||||||||||||||||
|
line 179-180: What do you mean. It offers the possibility to take action |
We corrected to – “Therefore, the novel aspect is the application of reticulorumen boluses for real-time monitoring and lowering heat stress and it offers the possibility to take action in dairy cattle [24]” |
||||||||||||||||||||||||||||||||||||||||||||||||||||||||||||||||||||||||||||||||
|
line 238: estimated ? |
We corrected to – “We found high positive statistically significant relation between temperature humidity index and temperature…” |
||||||||||||||||||||||||||||||||||||||||||||||||||||||||||||||||||||||||||||||||
|
line 266-268: were are the diurnal data?
|
We corrected and added new section – “discussion” |
Reviewer 3 Report
The topic of the article is interesting, however, it lacks the indication, I believe that due to forgetfulness of the entire statistical methodology of the work, it is mandatory to put this information in the article.... For example, which tests are used, levels of significance, etc. ...
This article must have a major revision, including the above, to be accepted.
There should be greater uniformity of the text, for example, sometimes they use dots and other commas to separate the decimal places.
There are other points that must be changed indicated in the attached file.

Author Response
Dear Reviewer,
Authors are very thankful with the comments, which help us to improve the manuscript. All changes proposed have been included in the manuscript and highlighted in yellow and track changes.
Best Regards,
Prof. Ramunas Antanaitis
|
Question |
Answers |
||||||||||||||
|
The topic of the article is interesting, however, it lacks the indication, I believe that due to forgetfulness of the entire statistical methodology of the work, it is mandatory to put this information in the article.... For example, which tests are used, levels of significance, etc. ...
|
We have added information regarding statistical analysis.
Using descriptive statistics, normal distributions of variables were assessed using the Kolmogorov–Smirnov test. The results were produced as the mean standard error of the mean (M ± SE). Mean differences between groups were analysed using the Student’s t-test. The Pearson correlation coefficient was calculated to define the linear relationship between the investigated variables. A linear regression equation was calculated to determine the statistical relationship between methane CH4 (dependent variable) and date (independent variable) during each week (05/13, 06/08, 06/15, 06/22). A probability below 0.05 was considered reliable (p < 0.05). |
||||||||||||||
|
There should be greater uniformity of the text, for example, sometimes they use dots and other commas to separate the decimal places. |
Corrected |
||||||||||||||
|
General comment - The title does not seem appropriate to the study, I suggest you adapt the title with the article. They only related the stress caused by THI to acidosis, always highlighting the novelty of the work of using rumen boluses to control methane emission and the risk of stress. |
We corrected to –“Relationship Between Reticulorumen Parameters Measured in Real Time and Methane Emission and Heat Stress Risk in Dairy Cows” |
||||||||||||||
|
Suggestion: you should change the keywords since two of them are present in the title. |
We corrected to –“ precision farming; green gas haus emmision; global warming“ |
||||||||||||||
|
Abstract In the abstract they should be clearer and indicate the methodology used, make somebreferences, but it is not clear. They should also indicate more concretely the results obtained. We must read the abstract and understand all the work, in this summary the objective of the work and the conclusion are well defined, the rest needs to be clearer. |
We corrected abstract and added information - “Feed contained a mixture of of grass silage (38%), corn silage (38%), flaked grain concentrate with mineral mixture (24 percent). Ingredients: 49% dry matter, 28% neutral detergent fiber, 20% acid detergent fiber, 16% crude protein, 39% non-fiber carbohydrates, and 39% net energy for lactation (Mcal/kg); net energy for lactation (Mcal/kg). According results of our study we found a relationship between reticulorumen parameters measured in real time and methane emission and heat stress risk in dairy cows.Cows assigned to the 2 nd pH class had higher (46,18 %) average values for methane emission, P < 0.01. While for the rest of indicators, the higher average values were detected in cows of 1 st pH class, RR temperature (2.80 %), relative humidity (20.96 %), temperature humidity index (2.47 %), (P < 0.01) and temperature (3.93%), (P < 0.05) higher compared to cows of 2 nd pH class. Reticulorumen pH was highly negatively related with THI and temperature (r=-0.667-0.717, P < 0.001) and medium negatively with heat index, relative humidity and RR temperature (r=-0.536, P < 0.001; r=-0.471 – 0.456, P < 0.01“ |
||||||||||||||
|
Introduction Only 5 references from the last 3 years (2 from 2021, 2 from 2020 and 1 from 2022 in the introduction). If, as they refer, it is a current topic, it should contain more current references. |
We added few new references –
“Methane is produced in the rumen as a result of manure degradation and microbial fermentation. Reduced CH4 emissions will halt climate change and lower greenhouse gas levels [7]. Increasing milk yield lowers climate consequences at the animal and farm levels [8]. The new findings could lead the way for more research into the effects of methanogenesis suppression on rumen fermentation and post-absorptive metabolism, which could increase animal productivity and efficiency [7]” “Future platforms for autonomous monitoring and mitigation of heat stress in cattle are likely to be based on minimally invasive smart technologies, either individually or as part of an integrated system, allowing for real-time solutions to animal responses in a variety of production systems and environmental conditions [16]” “Indeed, wireless telemetry technologies have been used to develop boluses for monitoring the pH of the rumen, and are now frequently used to control/measure physiological parameters that can indicate livestock diseases, such as the detection of subacute ruminal acidosis and other variations in the ruminal environment. As a result, this technique increases illness diagnosis timeliness and adds to the protection of animal health and productivity. Continuous rumen pH and temperature monitoring could be useful for assessing the impact of water temperature on ruminal parameters in cattle [16], [17]. At the moment, a bolus put into the rumen can measure a cow's temperature in real time (reticulorumen). The boluses can measure both temperature and pH. Wireless boluses can transmit data every ten minutes. The data can be saved in the cloud or on a PC. Measurements can be taken for up to a year depending on the battery life of the various bolus versions [18]. The sensor is positioned in the reticulorumen, where it is influenced by fermentation heat, which is 0.5 degrees Celsius greater than body temperature, and the transient cooling impact of the cow's drinking water [18]. Recent advancements in automated animal monitoring technology have showed promise in monitoring heat stress in cattle while taking individual behavioral and activity profiles into account [19]. Future platforms for autonomous monitoring and mitigation of heat stress in cattle are likely to be based on minimally invasive smart technologies, either individually or as part of an integrated system, allowing for real-time solutions to animal responses in a variety of production systems and environmental conditions [19]”
|
||||||||||||||
|
L125 is repeated “net energy for lactation (Mcal/kg).” |
We deleted repetition of “net energy for lactation (Mcal/kg).” |
||||||||||||||
|
The chemical composition of the feed should be presented in a table to make it more understandable. |
We added table with chemical composition of feed ration – Table 1. Chemical composition of feed ration.
|
||||||||||||||
|
Has any other author used smaXtec boluses? |
We corrected to – “It was possible to monitor real-time parameters such as pH, temperature of reticulorumen content (TRR), and cow activity by smaXtec boluses (smaXtec animal care technology®) [16], [17]” |
||||||||||||||
|
Are there references to laser use (LMD) in other articles for measuring methane? |
We added information and references – “The laser methane detector was designed to detect gas leaks from a safe distance in gas transmission networks, landfills, and other sites where there is a possibility of CH4 leakage [26]. Chagunda et al. [26] were the first to use the LMD to measure the concentration of CH4 in dairy cow breath. The LMD has various advantages, including its flexibility, portability, and ease of use. It also does not require an external power supply. As a result, it is reasonably inexpensive to employ in a wide range of experimental and commercial situations [27]. Researchers have further developed and evaluated the measurement, refined the analysis of data obtained with the LMD, and applied the LMD in studies on genetic analyses [28], [29], nutrition and feed efficiency [30], [31], the physiological status of animals [29], and to characterize different husbandry systems [32]. Chagunda et al.[26] found sensitivity and specificity of 95.4% and 96.5% for cows, respectively, and sensitivity and specificity of 93.8% and 78.7% for sheep. The measurement with the handheld LMD HESAI HS4000 (Hesai, Building L2-B, Hongqiao World Centre, Shanghai) is based on infrared absorption spectroscopy. By detecting a fraction of the diffusely reflected laser beam, the integrated CH4 concentration between the LMD and the target is determined [33]. The measured value is represented as CH4 column density (ppm), which is the sum of the CH4 concentrations along the laser route or the average CH4 concentration (ppm) multiplied by the path length (m). The LMD measures CH4 from 0.5 to 50,000 ppm m (up to 5 vol%) and can be utilized from a distance of 0.5 to 30 m. The data is displayed in real time on the LMD's display, and an audio and visual warning is issued if a particular threshold is surpassed. Following the physiology of the cow, the gas excreted directly from the rumen (eructation) is first inhaled into the lungs and then exhaled again with each respiratory cycle. LMD is aimed at the area around the animal’s nostrils, which is the main point source of emitted CH4. The measurements are performed by the same operator during the study period. An operator holds the LMD by hand and follows the animal’s head movements. Chagunda et al. [26] were the first to use the LMD to measure the concentration of CH4 in dairy cow breath. They used the LMD at a distance of 3 m from the cow and recorded for 15-25 seconds at a time at the nostrils. Distance of 3 m from the cow and taking recordings at the nostrils for 15–25 s at a time. Measurement intervals 0,5-1 s (i.e., one or two CH4 values per second) [34]. Measurements performed in an experiment at the same time of day – 2 hours after feeding. During the study period all the measurements performed same time of the day, approaching cows in the most similar way, same distance and keeping same LMD angle [27]”
|
||||||||||||||
|
The point regarding statistical analysis is omitted, it only indicates that you used SPSS to perform the statistical analysis…. All information is missing about the statistical analysis, which tests were used, how significance was obtained, etc. |
We added information in methods and materials section - “2.4. Data Analysis and Statistics. For the statistical analysis, SPSS 25.0 (IBM Corp., software.) Version 25.0 of IBM SPSS Statistics for Windows from IBM Corp. in Armonk, New York, the United States. Using descriptive statistics, normal distributions of variables were assessed using the Kolmogorov–Smirnov test. The results were produced as the mean standard error of the mean (M ± SE). Mean differences between groups were analysed using the Student’s t-test. The Pearson correlation coefficient was calculated to define the linear relationship between the investigated variables”
|
||||||||||||||
|
What correlations did you use? None of this is described in the work and it should be. Is required. |
We have added an information regarding this question.
The Pearson correlation coefficient was calculated to define the linear relationship between the investigated variables. |
||||||||||||||
|
Point 2.1.1.4 should be included in the experimental design. |
We included this point into the experimental design. |
||||||||||||||
|
How can they make reference to statistical differences in results and discussion if they don't indicate how they got there? |
We added information about statistical methods. |
||||||||||||||
|
Results and discussion should be improved and more explicit. |
We corrected and added new section “discussion” |
||||||||||||||
|
The Results presented must be the same in their formatting. There are tables that the decimal place is separated with “commas” others with “dots”. Authors should review this entire chapter. |
Corrected table 3 and in whole manuscript. |
||||||||||||||
|
References L306- the date must be in bold |
Corrected |
Reviewer 4 Report
This manuscript describes a post-hoc analysis of dairy cows having low or high ruminal pH on methane emissions and relation with heat stress indicators. First, the manuscript is very poorly written. Some places are difficult to understand and there is inadequate information to understand what was done. This manuscript needs a lot of improvement before publication.
Second, there is inadequate explanation of data collection and analysis to understand what was done. And, erratic methane emissions measured 3 times on 4 days for only 15-25 seconds with a handheld laser raises concerns about accuracy and precision. No validation of this method was presented.
Specific comments:
L19-26 - the simple summary needs to be expanded and rewritten to better explain what was done and the significance. Lay person does not know what on-line system is.
L25 - how do boluses to measure pH control methane emissions? the word control is used several times. I do not think control is the correct word.
L36 - include the description of the diet fed when pH was measured. no pH value is reported for group 2. The previous paper [17] reports 4 groups instead of 2 and does not describe how the pH values for separating groups was decided.
L40-44 - section needs to provide more data to support conclusion. I disagree that rumen pH boluses are novel and again do not control methane emissions
L47 - the introduction could be more concise
L76-77 - these 2 sentences contradict each other. methane stopped at 6.0, but then started again at 6.0
L89-93 - this section is repeated from above
L95 - what is a significant-yielding cow? I do not think significant is the correct word
L120 - the diet composition only adds to 60%
L125 - what is 39% net energy for lactation? 'net energy for lactation (Mcal/kg)' is repeated.
Table 1 - instead of using 'value' for units, I suggest either a dash (-) or 'unitless'
L154 - I am not aware of data indicating that CH4 is absorbed across the rumen into the bloodstream
L160-165 - citations for validation of this methodology to accurately and precisely measure daily methane emissions is needed
L167 - description of data analysis is woefully inadequate. what statistical model was used? what factors were included in the model? what method was used to compute correlation coefficients?
L170-171 - much more description of separation of cows into 2 pH groups is needed. No pH value is given for group 2. Are these pH values the average of 24 hours for how many days? Reference 17 specifies 4 groups instead of 2 groups and does not explain how the groups were assigned.
Table 2 - how are there differences in heat index, relative humidity, temperature and THI between cows when cows were housed in the same facility at the same time? The only way I can see that this occurs is that this is really a day effect (different environemnt on different days of measurement). So does this mean that individual cows changed pH groups between measurement days based on their rumen pH on that day? There really has to be a lot more explanation of how measurements were taken (there is no mention of days until we get to Figure 4) and how cows were assigned to pH groups. We cannot really interpret the data because we cannot understand how it was collected and analyzed.
L197-208 - there is no reference to your own data in this discussion
L210-211 - I suggest using the word 'to' in the paretheses. (r=-0.667 to -0.717, P < 0.001)
L213 - you did not measure lactate in this study
L212-226 - there is no connection back to your data in this section
L239 - I think it should be 'index and temperature'
L255 - please change to '11 a.m. on 13th of June'
L256 - please change to '58.67% lower than 22nd of June' and '57.64% lower than 8th of June'
Figure 4 - the superscripts are really confusing. My interpretation of the data - every date at 9 h should have a capital letter (13th and 22nd are different but 8th and 15th are not different from 13th or 22nd and so 8th and 15th should have capital AD), 11 ho n 13th should only have 1 capital letter, and - 9 h on 13th should have one lowercase letter not 2 letters. It seems like you are trying to show multiple levels of significance using multiple asterisks. Pick 1 significance level for each comparison - either significant at 0.05 or 0.01. In the caption P < 0.05 is given twice.
L264 - you did not report diurnal ruminal pH
L264-281 - there is no reference back to your data to compare with previous results
L287 - the word 'controlling' is used again
Author Response
Dear Reviewer,
Authors are very thankful with the comments, which help us to improve the manuscript. All changes proposed have been included in the manuscript and highlighted in yellow and track changes.
Best Regards,
Prof. Ramunas Antanaitis
|
Question |
Answers |
|||||||||||||||||||||||||
|
This manuscript describes a post-hoc analysis of dairy cows having low or high ruminal pH on methane emissions and relation with heat stress indicators. First, the manuscript is very poorly written. Some places are difficult to understand and there is inadequate information to understand what was done. This manuscript needs a lot of improvement before publication.
|
We have made corrections also a statistical analysis. |
|||||||||||||||||||||||||
|
Second, there is inadequate explanation of data collection and analysis to understand what was done. And, erratic methane emissions measured 3 times on 4 days for only 15-25 seconds with a handheld laser raises concerns about accuracy and precision. No validation of this method was presented. |
We added information – “We added information and references – “The laser methane detector was designed to detect gas leaks from a safe distance in gas transmission networks, landfills, and other sites where there is a possibility of CH4 leakage [26]. Chagunda et al. [26] were the first to use the LMD to measure the concentration of CH4 in dairy cow breath. The LMD has various advantages, including its flexibility, portability, and ease of use. It also does not require an external power supply. As a result, it is reasonably inexpensive to employ in a wide range of experimental and commercial situations [27]. Researchers have further developed and evaluated the measurement, refined the analysis of data obtained with the LMD, and applied the LMD in studies on genetic analyses [28], [29], nutrition and feed efficiency [30], [31], the physiological status of animals [29], and to characterize different husbandry systems [32]. Chagunda et al.[26] found sensitivity and specificity of 95.4% and 96.5% for cows, respectively, and sensitivity and specificity of 93.8% and 78.7% for sheep. The measurement with the handheld LMD HESAI HS4000 (Hesai, Building L2-B, Hongqiao World Centre, Shanghai) is based on infrared absorption spectroscopy. By detecting a fraction of the diffusely reflected laser beam, the integrated CH4 concentration between the LMD and the target is determined [33]. The measured value is represented as CH4 column density (ppm), which is the sum of the CH4 concentrations along the laser route or the average CH4 concentration (ppm) multiplied by the path length (m). The LMD measures CH4 from 0.5 to 50,000 ppm m (up to 5 vol%) and can be utilized from a distance of 0.5 to 30 m. The data is displayed in real time on the LMD's display, and an audio and visual warning is issued if a particular threshold is surpassed. Following the physiology of the cow, the gas excreted directly from the rumen (eructation) is first inhaled into the lungs and then exhaled again with each respiratory cycle. LMD is aimed at the area around the animal’s nostrils, which is the main point source of emitted CH4. The measurements are performed by the same operator during the study period. An operator holds the LMD by hand and follows the animal’s head movements. Chagunda et al. [26] were the first to use the LMD to measure the concentration of CH4 in dairy cow breath. They used the LMD at a distance of 3 m from the cow and recorded for 15-25 seconds at a time at the nostrils. Distance of 3 m from the cow and taking recordings at the nostrils for 15–25 s at a time. Measurement intervals 0,5-1 s (i.e., one or two CH4 values per second) [34]. Measurements performed in an experiment at the same time of day – 2 hours after feeding. During the study period all the measurements performed same time of the day, approaching cows in the most similar way, same distance and keeping same LMD angle [27]” We have added information regarding data Analysis and statistics. Using descriptive statistics, normal distributions of variables were assessed using the Kolmogorov–Smirnov test. The results were produced as the mean standard error of the mean (M ± SE). Mean differences between groups were analysed using the Stu-dent’s t-test. The Pearson correlation coefficient was calculated to define the linear relationship between the investigated variables. A linear regression equation was calculated to determine the statistical relationship between methane CH4 (dependent variable) and date (independent variable) during each week (05/13, 06/08, 06/15, 06/22). The probability below 0.05 was considered reliable (p < 0.05). |
|||||||||||||||||||||||||
|
Specific comments: |
|
|||||||||||||||||||||||||
|
L19-26 - the simple summary needs to be expanded and rewritten to better explain what was done and the significance. Lay person does not know what on-line system is. |
We corrected simple summary – “In literature we found only a few publications about rumen pH and CH4 connection. Based on this we hypothesized that reticulorumen pH and temperature, registered with on line system has impact on green gas CH4 emission. As a result, to the best of our knowledge, this is the first study that assesses the relationship between CH4 emission and reticulorumen pH and temperature. According to the study's goal, we found that cows with the higher pH (6.22–6.42) produce 46.18% more methane emissions than cows with the lower pH. Also, cows with a higher risk of heat stress had a higher risk of subclinical acidosis. The novel aspect is that by using real-time reticulorume pH, temperature registration system, and the laser methane detector, we can find the relationship between reticulorumen parameters measured in real time and methane emission and heat stress risk in dairy cows. For this reason, more studies should be done to evaluate this process.”
|
|||||||||||||||||||||||||
|
L25 - how do boluses to measure pH control methane emissions? the word control is used several times. I do not think control is the correct word. |
We corrected to – “A novel aspect is to find a relationship between reticulorumen parameters measured in real time and methane emission and heat stress risk in dairy cows” |
|||||||||||||||||||||||||
|
L36 - include the description of the diet fed when pH was measured. no pH value is reported for group 2. The previous paper [17] reports 4 groups instead of 2 and does not describe how the pH values for separating groups was decided. |
We added – “Feed contained a mixture of of grass silage (38%), corn silage (38%), flaked grain concentrate with mineral mixture (24 percent).. Ingredients: 49% dry matter, 28% neutral detergent fiber, 20% acid detergent fiber, 16% crude protein, 39% non-fiber carbohydrates, and 39% net energy for lactation (Mcal/kg); net energy for lactation (Mcal/kg)”
Also we have added the amount of cows for each pH group. The experimental animals were separated into two groups based on the reticulorumen pH assay: 1. pH< 6.22 (n=25, 69.0% of cows), 2. pH 6.22–6.42 (n=11, 31.0% of cows). Classes were assigned according to our previous publication [22].
|
|||||||||||||||||||||||||
|
L40-44 - section needs to provide more data to support conclusion. I disagree that rumen pH boluses are novel and again do not control methane emissions |
We corrected to – “Feed contained a mixture of grass silage, flaked grain concentrate (50 percent), grass hay (5 percent), and a mineral mixture (5 percent). Ingredients: 49% dry matter, 28% neutral detergent fiber, 20% acid detergent fiber, 16% crude protein, 39% non-fiber carbohydrates, and 39% net energy for lactation (Mcal/kg); net energy for lactation (Mcal/kg). According results of our study we found a relationship between reticulorumen parameters measured in real time and methane emission and heat stress risk in dairy cows.Cows assigned to the 2 nd pH class had higher (46,18 %) average values for methane emission, P < 0.01. While for the rest of indicators, the higher average values were detected in cows of 1 st pH class, RR temperature (2.80 %), relative humidity (20.96 %), temperature humidity index (2.47 %), (P < 0.01) and temperature (3.93%), (P < 0.05) higher compared to cows of 2 nd pH class. Reticulorumen pH was highly negatively related with THI and temperature (r=-0.667-0.717, P < 0.001) and medium negatively with heat index, relative humidity and RR temperature (r=-0.536, P < 0.001; r=-0.471 – 0.456, P < 0.01. |
|||||||||||||||||||||||||
|
L47 - the introduction could be more concise
|
We corrected introduction and according recommendations of other reviewer we added information about novelty about boluses and CH4 |
|||||||||||||||||||||||||
|
L76-77 - these 2 sentences contradict each other. methane stopped at 6.0, but then started again at 6.0 |
We corrected to – “Van Kessel and Russell [11] discovered that when the pH of rumen fluid from forage-fed cows was decreased to 6.0, in vitro CH4 generation stopped” |
|||||||||||||||||||||||||
|
L89-93 - this section is repeated from above |
We deleted this sentence. |
|||||||||||||||||||||||||
|
L95 - what is a significant-yielding cow? I do not think significant is the correct word |
We corrected to – “High yielding Holstein…” |
|||||||||||||||||||||||||
|
L120 - the diet composition only adds to 60%
|
We corrected to – “TMR contained a mixture of grass silage (38%), corn silage (38%), flaked grain concentrate with mineral mixture (24 percent)” |
|||||||||||||||||||||||||
|
L125 - what is 39% net energy for lactation? 'net energy for lactation (Mcal/kg)' is repeated. |
Deleted repetition. |
|||||||||||||||||||||||||
|
Table 1 - instead of using 'value' for units, I suggest either a dash (-) or 'unitless' |
Corrected to – dash (-) |
|||||||||||||||||||||||||
|
L154 - I am not aware of data indicating that CH4 is absorbed across the rumen into the bloodstream |
We deleted this sentence. |
|||||||||||||||||||||||||
|
L160-165 - citations for validation of this methodology to accurately and precisely measure daily methane emissions is needed |
We added – “..During the study period all the measurements performed same time of the day, approaching cows in the most similar way, same distance and keeping same LMD angle [20]” |
|||||||||||||||||||||||||
|
L167 - description of data analysis is woefully inadequate. what statistical model was used? what factors were included in the model? what method was used to compute correlation coefficients? |
We added information – “Using descriptive statistics, normal distributions of variables were assessed using the Kolmogorov–Smirnov test. The results were produced as the mean standard error of the mean (M ± SE). Mean differences between groups were analysed using the Student’s t-test. The Pearson correlation coeficient was calculated to define the linear relationship between investigated variables”
|
|||||||||||||||||||||||||
|
L170-171 - much more description of separation of cows into 2 pH groups is needed. No pH value is given for group 2. Are these pH values the average of 24 hours for how many days? Reference 17 specifies 4 groups instead of 2 groups and does not explain how the groups were assigned. |
We added – “…pH 6.22–6.42 (31.0% of cows…” We added - The experimental animals were separated into two groups based on the reticulorumen pH assay: 1. pH< 6.22 (n=25, 69.0% of cows), 2. pH 6.22–6.42 (n=11, 31.0% of cows). It was possible to monitor real-time parameters such as pH, temperature of reticulorumen content (TRR), and cow activity by smaXtec boluses (smaXtec animal care technology®) [24], [25]. All methodology of monitoring of real time parameters is added in a manuscript, but we also added this information. The data from reticulorumen were recorded at the same intervals as other investigated traits: 05/13, 06/08, 06/15 and 06/22 . |
|||||||||||||||||||||||||
|
Table 2 - how are there differences in heat index, relative humidity, temperature and THI between cows when cows were housed in the same facility at the same time? The only way I can see that this occurs is that this is really a day effect (different environemnt on different days of measurement). So does this mean that individual cows changed pH groups between measurement days based on their rumen pH on that day? There really has to be a lot more explanation of how measurements were taken (there is no mention of days until we get to Figure 4) and how cows were assigned to pH groups. We cannot really interpret the data because we cannot understand how it was collected and analyzed. |
We added in materials and methods – “The experiment was carried out at the Lithuanian University of Health Sciences, with 650 milking Holstein cows (55.792368°N, 24.017499° E) in one of Lithuania's dairy farms from April to August 2022. The study was conducted on clinically healthy cows in their second lactation (that had an average daily milk yield of 32.19 ± 1.05 kg per cow), average of feed intake - 18 kg DM/day, milk fat – 4.25 (±0.25) milk protein – 3.45 (±0.15), milk somatic cell count -180000/ml (±0.55), milk urea nitrogen – 25% (±5). Cows were kept in a free stall barn and milked using a Delaval milking parlour”
“The two groups of animals were created on the basis of registered parameters from April to August 2022. During this period, conditions changed only keeping condition (heat index, relative humidity, temperature, and THI)”
Also, we corrected – Table 2. Parameters and intervals of their measurements.
|
|||||||||||||||||||||||||
|
L197-208 - there is no reference to your own data in this discussion |
We added –“[13]. Results of our research revealed that cows assigned to the 2 nd pH class had higher (46,18 %) average values for methane emission (Table 3)” |
|||||||||||||||||||||||||
|
L210-211 - I suggest using the word 'to' in the paretheses. (r=-0.667 to -0.717, P < 0.001) |
We corrected to – “Reticulorumen pH was highly negatively related with THI and temperature (r=-0.667 to 0.717, P < 0.001) and medium negatively with heat index, relative humidity and RR temperature (r=-0.536, P < 0.001; r=-0.471 to 0.456, P < 0.01), (Figure 1)” |
|||||||||||||||||||||||||
|
L213 - you did not measure lactate in this study |
We deleted this sentence. |
|||||||||||||||||||||||||
|
L212-226 - there is no connection back to your data in this section |
We deleted this sentence. |
|||||||||||||||||||||||||
|
L239 - I think it should be 'index and temperature' |
We corrected to – “…temperature humidity index and temperature…” |
|||||||||||||||||||||||||
|
L255 - please change to '11 a.m. on 13th of June' |
Corrected to – “The lowest methane emission was detected at 11 a.m. on 13th of June 58.79% lower compared to 15 th of June (P < 0.01), 58.67% lower than 22nd of June (P < 0.05) and '57.64% lower than 8th of June (P < 0.001), (Figure 4)”
|
|||||||||||||||||||||||||
|
L256 - please change to '58.67% lower than 22nd of June' and '57.64% lower than 8th of June' |
Corrected to – “The lowest methane emission was detected at 11 a.m. on 13th of June 58.79% lower compared to 15 th of June (P < 0.01), 58.67% lower than 22nd of June (P < 0.05) and '57.64% lower than 8th of June (P < 0.001), (Figure 4)”
|
|||||||||||||||||||||||||
|
Figure 4 - the superscripts are really confusing. My interpretation of the data - every date at 9 h should have a capital letter (13th and 22nd are different but 8th and 15th are not different from 13th or 22nd and so 8th and 15th should have capital AD), 11 ho n 13th should only have 1 capital letter, and - 9 h on 13th should have one lowercase letter not 2 letters. It seems like you are trying to show multiple levels of significance using multiple asterisks. Pick 1 significance level for each comparison - either significant at 0.05 or 0.01. In the caption P < 0.05 is given twice. |
We have changed to a table, for a better representation of the data.
Table 4. Distribution of methane emission by hour and date. Different lower letters a, b, c indicate statistically significant differences between hours; Capital letter A, B, C, D indicate statistically significant differences between date. * P < 0.05, ** P < 0.05, ** P < 0.01, *** P < 0.001). Changed from figure 4 to table 4. Lower letters a, b, c indicate statistically significant differences between hours, while capital letter A, B, C, D indicate statistically significant differences between date. |
|||||||||||||||||||||||||
|
L264 - you did not report diurnal ruminal pH |
We corrected and added new discussion section. |
|||||||||||||||||||||||||
|
L264-281 - there is no reference back to your data to compare with previous results |
We corrected and added new discussion section |
|||||||||||||||||||||||||
|
L287 - the word 'controlling' is used again |
We corrected to – “The novel aspect is found the relationship between reticulorumen parameters measured in real time and methane emission and heat stress risk in dairy cows“ |
Round 2
Reviewer 3 Report
The abstract is too long, it does not comply with what is stipulated by the journal (200 words).
The units in table 1 are not complete. To be expressed in %, they should refer to dry matter. Check the units.
It has repeated information L197 and L221, must correct the text to avoid repetitions.
Author Response
Dear Reviewer,
Authors are very thankful with the comments, which help us to improve the manuscript. All changes proposed have been included in the manuscript and highlighted in yellow and track changes.
Best Regards,
Prof. Ramunas Antanaitis
|
Question |
Answers |
||||||||||||
|
The abstract is too long, it does not comply with what is stipulated by the journal (200 words). |
We corrected abstract section – “The objective of the study was to investigate a connection between CH4 emissions and reticulorumen pH and temperature. During experiment we registered following parameters: reticulorumen pH (pH), reticulorumen temperature (RR temp.), reticulorumen temperature without drinking cycles, ambient temperature, ambient relative humidity, cow activity, heat index, temperature – humidity index (THI) and methane emission (CH4). The experimental animals were divided into two groups based on the reticulorumen pH: 1. pH <6.22, 2. pH 6.22–6.42. We found that cows assigned to the 2 nd pH class had higher (46,18 %) average values for methane emission, P < 0.01. While for the rest of indicators, the higher average values were detected in cows of 1 st pH class, RR temperature (2.80 %), relative humidity (20.96 %), temperature humidity index (2.47 %), (P < 0.01) and temperature (3.93%), (P < 0.05) higher compared to cows of 2 nd pH class. Reticulorumen pH was highly negatively related with THI and temperature (r=-0.667-0.717, P < 0.001) and medium negatively with heat index, relative humidity and RR temperature (r=-0.536, P < 0.001; r=-0.471 – 0.456, P < 0.01.Cows with higher risk of heat stress had higher risk of lower reticulorumen pH” |
||||||||||||
|
The units in table 1 are not complete. To be expressed in %, they should refer to dry matter. Check the units. |
We corrected table 1 – Table 1. Chemical composition of feed ration.
DM—dry matter; % of DM—percent of dry matter; Mcal/kg—megacalorie per kilogram. |
||||||||||||
|
It has repeated information L197 and L221, must correct the text to avoid repetitions. |
We deleted repetition |
Reviewer 4 Report
The authors have made significant improvements to the manuscript, but there are still some things to work on. The english grammar still needs more work.
L50 - these are not ingredients. they are nutrients
L59,60 - use the word to instead of a dash between numbers
L77 - methane in the rumen is not a result of manure degradation. I think some wording needs changed in this sentence
L229 - my interpretation of the statistical analysis is that a student T-test was performed comparing the 2 pH groups, then a T-test comparing the days (which would actually require multiple T-test since more than 2 categories). There is lots of issues with this analysis. You are assuming that pH group and day are independent which they might not be. And you not accounting for correlation between repeated measurments on the same cow at 4 time intervals. The stats analysis needs to be redone using a repeated measures model with pH group, week, time of day, group x week, group X time, time x week, and group x time x week interactions with cow as the subject for repeated measures.
L350-362 - how is your measurement of pH different because you have a clear difference in methane emissions of low and high pH groups in vivo?
L436 - it sounds like methane emissions is influenced by a combination of pH, temperature, and heat stress. Would a multivariate regression model be appropriate to understand how the combination of these factors influence methane emissions? This paper evaluates each factor separately, whereas it seems that a combination is more appropriate.
Author Response
Dear Reviewer,
Authors are very thankful with the comments, which help us to improve the manuscript. All changes proposed have been included in the manuscript and highlighted in yellow and track changes.
Best Regards,
Prof. Ramunas Antanaitis
|
Question |
Answers |
||||||||||||||||||||||||||||||||||||||||||
|
The authors have made significant improvements to the manuscript, but there are still some things to work on. The english grammar still needs more work. |
We corrected in whole manuscript
|
||||||||||||||||||||||||||||||||||||||||||
|
L50 - these are not ingredients. they are nutrients |
We deleted this sentence from abstract section and added. We corrected in methodological section (table 1). |
||||||||||||||||||||||||||||||||||||||||||
|
L59,60 - use the word to instead of a dash between numbers
|
We corrected to – “Reticulorumen pH was highly negatively related with THI and temperature (r=-0.667 to 0.717, P < 0.001) and medium negatively with heat index, relative humidity and RR temperature (r=-0.536, P < 0.001; r=-0.471 to 0.456, P < 0.01. Cows with higher risk of heat stress had higher risk of lower reticulorumen pH” |
||||||||||||||||||||||||||||||||||||||||||
|
L77 - methane in the rumen is not a result of manure degradation. I think some wording needs changed in this sentence
|
We corrected to – “Methane is produced in the rumen as a result of microbial fermentation” |
||||||||||||||||||||||||||||||||||||||||||
|
L229 - my interpretation of the statistical analysis is that a student T-test was performed comparing the 2 pH groups, then a T-test comparing the days (which would actually require multiple T-test since more than 2 categories). There is lots of issues with this analysis. You are assuming that pH group and day are independent which they might not be. And you not accounting for correlation between repeated measurments on the same cow at 4 time intervals. The stats analysis needs to be redone using a repeated measures model with pH group, week, time of day, group x week, group X time, time x week, and group x time x week interactions with cow as the subject for repeated measures.
|
Due to that ph, heat index, was measured 1 time per month, there is only records of 1 cow, while a time and week record there measured not one time. For example we don’t have a value for 9, then 10 and then 11 a.m. it is one value on that day… so there is no possibility to calculate ph, temperature and heat index with repeated measures as CH4 at different hours, or different weeks. We are sending a calculation for CH4 repeated measures.
|
||||||||||||||||||||||||||||||||||||||||||
|
L350-362 - how is your measurement of pH different because you have a clear difference in methane emissions of low and high pH groups in vivo?
|
Our research was different because it used a real-time reticulorumen-measured system. We corrected and added this information – “….In our research, using real-time measured reticulorumen parameters, we found that cows with a reticulorumen pH of 6.22–6.42 had 46.18% higher average values for methane emission”
|
||||||||||||||||||||||||||||||||||||||||||
|
L436 - it sounds like methane emissions is influenced by a combination of pH, temperature, and heat stress. Would a multivariate regression model be appropriate to understand how the combination of these factors influence methane emissions? This paper evaluates each factor separately, whereas it seems that a combination is more appropriate. |
Multivariate linear regression
CH4= - 5692,182 + (6.777 x heat index) + (116.055 x temperature) + (637.882 x RR pH). R= 0.560, R square = 0.314
Multivariate regression analysis showed that then heat index is increasing in one unit methane is also increasing by 6.777 ppm (P>0.05); then temperature is increasing in 1 °C, methane emission is also increasing by 116,055ppm (P<0.05), then RR ph is increasing in one unit – methane is increasing also by 637.882ppm (P<0.01).
|
||||||||||||||||||||||||||||||||||||||||||